# Structural Characterization of a *Pleurotus sajor-caju* Polysaccharide and Its Neuroprotection Related to the Inhibition of Oxidative Stress

**DOI:** 10.3390/nu14194047

**Published:** 2022-09-29

**Authors:** Mengdi Liu, Yingdi Dai, Chengming Song, Jia Wang, Yang Liu, Qi Wang

**Affiliations:** 1Engineering Research Center of Chinese Ministry of Education for Edible and Medicinal Fungi, Jilin Agricultural University, Changchun 130118, China; 2College of Plant Protection, Jilin Agricultural University, Changchun 130118, China; 3Guang’anmen Hospital China Academy of Chinese Medical Sciences Respiratory Department, Beijing 100053, China

**Keywords:** *Pleurotus sajor-caju*, polysaccharide, structural characterization, neuroprotection

## Abstract

A novel polysaccharide PSP2-1 was isolated and purified from *Pleurotus sajor-caju*. The structural characterization data displayed that the molecular weight of PSP2-1 was 44.9 kDa, and PSP2-1 consisted of fucose, galactose, glucose, and mannose. The methylation results showed that the glycosidic bonds of PSP2-1 included T-Fuc, 1,6-Gal, T-Glc, 1,6-Glc, 1,3,6-Glc, 1,3-Man, 1,2,6-Man, and T-Man. Neuroprotective studies indicated that PSP2-1 significantly improved the cell viability of the H_2_O_2_-induced oxidatively damaged neuronal cell HT22, reduced the release of LDH, inhibited apoptosis and release of cytochrome c, and alleviated the decline of mitochondrial membrane potential and ROS accumulation. Furthermore, PSP2-1 decreased the phosphorylation levels of cleaved PARP and cleaved caspase-3, and increased the ratio of bcl-2/bax. Additionally, PSP2-1 could inhibit the phosphorylation of MAPK family members including JNK, p38, and Erk. Finally, animal experiments showed that PSP2-1 could improve the oxidative stress injury and the learning and memory ability of mice with aging induced by D-galactose. Our results confirmed that PSP2-1 significantly ameliorated the oxidative stress injury, inhibited the apoptosis in H_2_O_2_-induced neuronal cells via MAPK pathway, and also improved cognition in mice with aging induced by D-galactose. Our research gives the foundation for the functional food application of *P. sajor-caju* polysaccharides in the future.

## 1. Introduction

Oxidative stress (OS) is a series of cascade reactions mediated by the overproduction of reactive oxygen species (ROS) [1]. Excessive ROS, including superoxide anion, hydrogen peroxide (H_2_O_2_), and hydroxyl radicals, can lead to mitochondrial dysfunction, apoptosis, and DNA damage of nerve cells [2]. Therefore, OS is an essential pathogenic factor leading to neurodegenerative diseases, such as aging Alzheimer’s disease (AD), Parkinson’s disease (PD), etc. [3,4,5]. An effective method for the prevention and treatment of neurodegenerative diseases is expected to be discovered by exploring active substances to decrease the level of intracellular ROS and prevent oxidative injury [6].

Polysaccharides from fungi have been widely explored because of their antioxidant activities and few side effects [7]. Consequently, polysaccharides from mushrooms can be a potential drug to prevent oxidative damage of nerve cells and improve nervous system diseases. Several studies have been carried out and confirmed the *Pleurotus ostreatus* polysaccharide, *Hericium erinaceus* polysaccharide, and *Ganoderma lucidum* polysaccharide can protect nerve cells or improve the cognition of AD model mice [8,9,10]. *Pleurotus sajor-caju* belongs to basidiomycetes division, Agaricales order, and *Pleurotus* genus, and is commonly referred to as a “Houbitake” in Japan [11,12]. *P. sajor-caju* is widely distributed in China and very popular because of its delicious taste. Furthermore, the various biological activities of polysaccharides from *P. sajor-caju* have been reported, including anti-inflammatory, antioxidant, immunostimulatory, and anti-tumor activities [13,14,15,16]. However, the protective activity and possible mechanism of polysaccharides from *P. sajor-caju* on neuronal cells have not been studied.

In our previous study, we collected the strains of *P. sajor-caju* and successfully cultivated them in the laboratory to obtain fruiting bodies. In this paper, a new *P. sajor-caju* fruiting body polysaccharide was extracted and purified. Then, the physicochemical properties of the polysaccharide were determined by Fourier-transform infrared spectroscopy (FTIR), high performance anion exchange chromatography (HPAEC), gas chromatography–mass sepectrometry (GC-MS), and gel permeation chromatography–refractive index–multi-angle laser light scattering (GPC-RI-MALLS). In addition, we systematically investigated the neuro-protective and potential mechanism of the polysaccharide from *P. sajor-caju* against H_2_O_2_-induced neurotoxicity in HT22 cells, and verified the improvement of the oxidative stress injury and learning and memory ability of D-galactose-induced aging mice.

## 2. Results and Discussion

### 2.1. Purification and Physicochemical Characteristics of Polysaccharide PSP2-1

The polysaccharide from *P. sajor-caju* was obtained by water extraction and ethanol precipitation. Then, the purified polysaccharide was applied to a DEAE FF column, as shown in Figure 1A. The main polysaccharide fraction, named PSP2, was eluted by NaCl and pooled. Then, PSP2 was further purified by Sephacry S-400 HR column, as shown in Figure 1B; one fraction of PSP2-1 was obtained. As shown in Table 1, the carbohydrate and protein contents of PSP2-1 were 85.3% and 3.1%, respectively. The PSP2-1 was consisted of fucose, galactose, glucose, and mannose with mole percentages of 2.40%, 28.81%, 30.63%, and 37.79%, respectively (Figure 1C and Table 1). The monosaccharide composition of PSP2-1 is similar to the previous results reported by Cinthia B.S. Telles et al. [12], but it is quite different from a water soluble polysaccharide isolated from *Pleurotus sajor-caju*, cultivar Black Japan, which was only composed of D-glucose and D-galacotose (3:1) [17]. These differences in monosaccharide ratios may be due to several factors, including extraction, purification methods, and the differences of mushroom source. UV spectrum of PSP2-1 showed no absorption at 260 nm and 280 nm, suggesting that there was no or little nucleic acid and protein in PSP2-1 (Figure 1D). Furthermore, the FTIR spectrum of PSP2-1 was determined, as shown in Figure 1E. The broad peak of 3392 cm^−1^ was assigned to O-H stretching vibrations [18]. The absorption peak at 2933 cm^−1^ represented the C-H stretching [19]. The band at 1647 cm^−1^ was due to the carbonyl bond stretching vibration [20]. The peak at 1416 cm^−1^ was attributed to the C-H deformation vibration [21]. The peaks at 800–1200 cm^−1^ represented the area of carbohydrates [22]. The singles at 1148 cm^−1^ and 1079 cm^−1^ were due to the stretch vibration of C-O-C linkages and the pyranoid ring [23]. The bands at 871 cm^−1^ and 799 cm^−1^ indicated the existence of a configuration of α-glycosides [24]. In addition, the molecular weight of PSP2-1 was determined by GPC-RI-MALS. Figure 1E and Table 2 showed that the molecular weight of PSP2-1 was 44.9 kDa.

### 2.2. Methylation Results of PSP2-1

Methylation detection is an important means to analyze the glycoside bond of polysaccharides. The methylated derivatives of PSP2-1 were determined by GC-MS. Table 2 showed that the PSP2-1 showed eight main components, namely, 1,5-di-O-acetyl-6-deoxy-2,3,4-tri-O-methyl fucitol, 1,5,6-tri-O-acetyl-2,3,4-tri-O-methyl galactitol, 1,5-di-O-acetyl-2,3,4,6-tetra-O-methyl glucitol, 1,5,6-tri-O-acetyl-2,3,4-tri-O-methyl glucitol, 1,3,5,6-tetra-O-acetyl-2,4-di-O-methyl glucitol, 1,3,5-tri-O-acetyl-2,4,6-tri-O-methyl mannitol, 1,2,5,6-tetra-O-acetyl-3,4-di-O-methyl mannitol, and 1,5-di-O-acetyl-2,3,4,6-tetra-O-methyl mannitol, with mole percentages of 1.51%, 25.57%, 19.57%, 4.33%, 6.74%, 13.44%, 16.72%, and 7.16%, respectively. The results suggested that the glycosidic bonds of PSP2-1 mainly included T-Fuc, 1,6-Gal, T-Glc, 1,6-Glc, 1,3,6-Glc, 1,3-Man, 1,2,6-Man, and T-Man. O-methyl galactans were in the main chain of *P. pulmonarius*, *P. eryngii*, and *P.ostreatoroseus* polysaccharides. These findings show that O-methyl-polysaccharides could be characteristic of the genus *Pleurotus* [25,26].

### 2.3. Effects of PSP2-1 on H_2_O_2_-Induced HT22 Cell

HT22 cells are mouse hippocampal neuron cell lines and have been widely studied as neuroprotective oxidative injury cell models. H_2_O_2_ is a drug that induces oxidative damage to nerve cells [27]. In this paper, the effects of PSP2-1 towards damage induced by H_2_O_2_ in HT22 cells were investigated. As shown in Figure 2A, there was no significant effect of PSP2-1 on HT22 cell viability in the range of 50–200 μg/mL, indicating that PSP2-1 had no cytotoxicity. As shown in Figure 2B, compared with the control, H_2_O_2_ significantly (*p* < 0.01) reduced the cell viability; then, PSP2-1 significantly increased the cell viability. Lactate dehydrogenase (LDH) is a glycolytic enzyme, which widely exists in the cytoplasm of all tissues or cells in the body [28,29]. When the cells are necrotic and apoptotic, the membrane structure is destroyed, which leads to the release of LDH from the cells. Therefore, the LDH release in the cell culture medium is an important indicator to reflect the degree of cell damage [30]. Figure 2C showed that the LDH release of model groups significantly (*p* < 0.05) increased, suggesting that the cell had damage; however, the PSP2-1 groups significantly reduced the LDH release. The results indicated that PSP2-1 had no cytotoxicity, and could alleviate the oxidative injury of nerve cells caused by H_2_O_2_.

### 2.4. PSP2-1 Inhibited H_2_O_2_-Induced HT22 Cell Mitochondrial Dysfunction

The DAPI staining cells were observed by fluorescence microscope, as shown in Figure 2D. The control group cells showed weak blue fluorescence and level dyeing, and, compared with control group, the H_2_O_2_-induced HT22 cell showed cell shrinkage, nuclear condensation, and bright blue fluorescence, indicating H_2_O_2_-induced HT22 cell apoptosis. However, the PSP2-1 effectively prevented the apoptosis induced by H_2_O_2_. Furthermore, the apoptosis rate was determined by the double-staining method with FITC-labeled annexin V and propidium iodide (PI). Figure 2E, F showed that the apoptosis rate of the model group was 17.8%, which was significantly higher than 2.1% of the control group. The apoptosis rate of cells treated with PSP2-1 significantly decreased, compared with the model group. This result was consistent with DAPI staining, which further confirmed that PSP2-1 could inhibit the apoptosis of H_2_O_2_-induced HT22 cell.

### 2.5. Effect of PSP2-1 on ROS Level

ROS-induced oxidative damage can lead to apoptosis, which is closely related to the pathological process of neurodegenerative diseases, such as AD and PD [31]. The accumulation of ROS and mitochondrial dysfunction are related to the upstream signal pathway induced by oxidative stress [32]. In this study, the effects of PSP2-1 on H_2_O_2_-induced ROS generation were investigated. The results, as shown in Figure 2G, H, demonstrate an increase in fluorescence intensity in the H_2_O_2_-treated HT22 cells compared with the control group; however, the *P. sajor-caju* polysaccharide PSP2-1 significantly decreased the fluorescence intensity. The results indicated that PSP2-1 could reduce the ROS level and alleviate oxidative damage in HT22 cells treated with H_2_O_2_.

### 2.6. Effect of PSP2-1 on the Mitochondrial Membrane and Release of Cytochrome c

As the main site of energy metabolism, the change of mitochondrial membrane potential is considered as an early marker of cell apoptosis. JC-1 is known to be a substrate of transporters and numerous fluorescent probes. Under normal physiological conditions, the mitochondrial membrane potential of JC-1 was higher, and JC-1 accumulated in the mitochondrial matrix to form a polymer and produced red fluorescence. When the cells were subjected to oxidative stress, the mitochondrial membrane potential decreased, and JC-1 could not gather in the mitochondrial matrix. At this time, JC-1 was a monomer and produced green fluorescence [33]. Previous studies found a good correlation between cellular accumulation of this dye and transporter expression levels, and much of the cellular outcome analysis was based on cellular accumulation of this dye [34]. Therefore, we determined the mitochondrial membrane potential of HT22 cells with JC-1 staining. As shown in Figure 3A, B, compared with the control group, green fluorescence intensity of nerve cells treated with H_2_O_2_ significantly increased. However, the green fluorescence intensity of cells treated with 50, 100, and 150 μg/mL of PSP2-1 significantly decreased, and the red fluorescence intensity markedly increased. The results suggested that H_2_O_2_ destroyed the mitochondrial membrane potential (MMP) of HT22 cells, resulting in mitochondrial dysfunction, and PSP2-1 could restore the mitochondrial dysfunction induced by H_2_O_2_.

The release of cytochrome c is an important marker of mitochondrial dysfunction, and also an important material for mitochondrial apoptosis [35]. Normally, cytochrome c is located outside the inner membrane of the mitochondria. After being stimulated by the apoptosis signal, cytochrome c is released and transferred into the cytoplasm. Cytochrome c binds to apoptotic protease activating factor-1 and oligomerizes to form a heptamer under the involvement of dATP [36]. Thus, the release of cytochrome c was examined by laser confocal microscopy. As shown in Figure 3C, in the control group, the cytochrome c is located in the mitochondria, showing a dot-like and dark green fluorescence, compared with the control group; after the HT22 cells were treated with H_2_O_2_, cytochrome c was released from the mitochondria to the cytoplasm, showing bright green fluorescence. However, the release of cytochrome c from the mitochondria to the cytoplasm was significantly alleviated. These results confirmed that PSP2-1 could prevent hydrogen peroxide-induced mitochondrial dysfunction in HT22 cells.

### 2.7. Effects of PSP2-1 on the Expression of Apoptotic Proteins in H_2_O_2_-Induced HT22 Cells

As everyone knows, the activation of caspase-3 can promote apoptosis signal transduction, cleave the death substrate PARP, and induce apoptosis [37]. In addition, bcl-2 family proteins also play a key role in the process of apoptosis. Bcl-2 is an anti-apoptotic protein and bax is a pro-apoptotic protein [38]. Bcl-2 plays a certain role in the survival of glial cells. Some researchers studied the hippocampal area of patients with nerve cell injury, and found that the expression of bcl-2 in damaged neurons was decreased, which indicated that brain cell damage in patients with a nerve cell injury was correlated with apoptosis to a certain extent [39]. When bcl-2 expression is upregulated, bax/bcl-2 dimer dissociates and is generated; then, bax/bcl-2 dimer decreases and inhibits apoptosis [40]. Thus, we investigated the expression of apoptotic-related proteins, such as cleaved caspase-3, cleaved PARP, bax, and bcl-2, with Western blot in HT22 cells after different treatments. As shown in Figure 4, compared with the control group, H_2_O_2_ treatment significantly increased the expression of cleaved caspase-3 and cleaved PARP, and decreased the expression ratio of bcl-2/bax. However, compared with the H_2_O_2_ treatment group, the PSP2-1 significantly decreased the expression of cleaved caspase-3 and cleaved PARP, and increased the expression ratio of bcl-2/bax. The results further indicated that PSP2-1 could protect nerve cells from oxidative damage by inhibiting hydrogen peroxide-induced cell apoptosis.

### 2.8. The Mechanism of PSP2-1 Inhibiting H_2_O_2_-Induced Apoptosis of Oxidatively Damaged Nerve Cells

To further elucidate the possible mechanism of PSP2-1 protecting HT22 nerve cells from oxidative damage induced by H_2_O_2_, we investigated the effect of PSP2-1 on the mitogen-activated protein kinase (MAPK) pathways by Western blot. The results are shown in Figure 5. The phosphorylation levels of Erk1/2, JNK, and p38 significantly increased in the H_2_O_2_ treatment group, compared with the control group. The results are consistent with the previous reports [41]. However, the expression levels of Erk1/2, JNK, and p38 gradually decreased after PSP2-1 treatment. MAPK, a serine/threonine protein kinase, including the three major subunits Erk1/2, JNK, and p38, is an important signal transduction system, involving cell differentiation, proliferation, apoptosis, and other cellular activities, which is closely related to neuronal oxidative damage [42]. The reactive oxygen species activation of the MAPK signaling pathway cascade is an important mechanism of oxidative stress-induced neuronal apoptosis [43]. MAPK is involved in most harmful abiotic stress processes, including oxidative stress, altered osmotic pressure, and DNA damage. Erk1/2, JNK, and p38 are involved in proliferation, differentiation, apoptosis, and stress response by regulating the activities of various proteins, enzymes, transcription factors, and key signaling pathways, and have a wide range of biological effects [44]. In this study, PSP2-1 can significantly inhibit the hydrogen peroxide-induced apoptosis of HT22 nerve cells and ROS production and alleviate mitochondrial dysfunction. In addition, PSP2-1 reduced the phosphorylation of Erk1/2, JNK, and p38 in HT22 cells that was induced by hydrogen peroxide. The results indicated that PSP2-1 could protect nerve cells by regulating the MAPK signaling pathway to inhibit oxidative damage and apoptosis induced by hydrogen peroxide (Figure 6).

### 2.9. Effects of PSP2-1 on Cognition of Mice with Aging Induced by D-Galactose

The Morris water maze (MWM) test is an effective means to detect the learning and memory ability of mice with neurodegenerative diseases [45,46]. D-galactose is a reducing sugar that can induce oxidative stress, leading to mitochondrial dynamics disorders and neuronal apoptosis [47]. D-galactose-induced brain aging in mice is similar to natural aging, which can lead to changes in cell osmotic pressure and accelerate neural degeneration, accompanied by increased oxidative stress [48,49]. To further confirm the pharmacological effect of the polysaccharide PSP2-1 on neurodegenerative diseases in animal models, we established an aging mouse model induced by D-galactose. Then, the swimming trace, escape latency time, and number of times to find the platform location when there is no platform were determined by MWM test. Figure 7A showed the thermal infrared trajectories of different groups of mice with and without platforms. The normal group of mice could almost directly find the resting platform, or when there is no platform, move to the platform area to find the removed platform. Regardless of whether there is a platform or not, the mice in the model group have no purpose to swim around irregularly, indicating that the mice have learning and memory disorders, and the polysaccharide group significantly proved this phenomenon. In addition, Figure 7B showed that mice in the normal group could find the platform in a short time, and the time for the model to find the platform is significantly (*p* < 0.5) higher than that of the control group. The time to find the platform in the polysaccharide group was significantly (*p* < 0.01) less than that of the model group. When there is no platform, the normal group of mice will go to the platform area many times to look for the removed platform. The number of mice in the model group looking for the platform is significantly (*p* < 0.01) lower than that of the control group; while compared with the model group, the number of mice in the polysaccharide group looking for the platform is significantly (*p* < 0.01) increased (Figure 7C). These results suggest that polysaccharides could improve the learning and memory ability of mice with aging induced by D-galactose. This is consistent with previous findings [50].

### 2.10. Effect of PSP2-1 on Oxidative Stress Injury Induced by D-Galactose in Aging Mice

To determine the degree of oxidative damage caused by D-galactose in aging mice, Malondialchehyche (MDA), Super Oxidase Dimutase (SOD), ROS, and Catalas (CAT) indexes in serum and brain tissues were determined. As shown in Figure 8, compared with the control group, the contents of MDA and ROS in the serum and brains of the model group were significantly increased, and the contents of CAT and SOD were decreased, which indicated that oxidative damage had occurred in aging mice (*p* < 0.01). After treatment with PSP2-1, compared with the model group, the contents of MDA and ROS the in serum and brains were significantly decreased, and the contents of CAT and SOD were increased in the PSP2-1 dose group, which indicated that PSP2-1 could improve oxidative damage in aging mice (*p* < 0.01). These oxidative index trends were similar to the results of polysaccharide activity studies of *Ganoderma*
*lucidum* previously published by Meng Jia et al. [51].

Many studies showed that the rational intake of typical plant-based eating patterns, such as the Mediterranean-style diet and some antioxidant/anti-inflammatory compounds, including COX/LOX modulators and isoflavones, could be used in the prevention of noncommunicable diseases, including neurodegenerative diseases [52,53]. Due to low toxicity and high pharmacological activity, *Pleurotus sajor-caju* polysaccharides, in combination with antioxidant compounds, could be added to nutraceuticals, which provides a possible further strategy in the therapy of neuroinflammatory diseases.

The present study in mice has several limitations. We only measured oxidative stress injury-related factors in the brains and serum of aging mice. However, we did not confirm through which pathway PSP2-1 decreased the oxidative stress injuries in mice. Additionally, aging is a time-dependent physiological process involving genomic instability, loss of proteostasis, deregulated nutrient sensing, intestinal microbiota changing, inflammation, and mitochondrial dysfunction. The underlying mechanism of PSP2-1 improving learning and memory ability in aging mice still needs to be further investigated.

## 3. Materials and Methods

### 3.1. Materials and Chemicals

The *P. sajor-caju* was collected in Hongyuan County, Sichuan Province, and the strain of the *P. sajor-caju* (T22002) was isolated and preserved in our laboratory. The strain was identified as *P. sajor-caju* by ITS sequencing (PCR, ITS sequence alignment results, and cultivation pictures are provided in the Appendix A). The fruit body was successfully cultivated by our laboratory. Fetal bovine serum (FBS), Dulbecco’s modified Eagle’s medium (DMEM), Biomyc-3, penicillin-streptomycin, trypsin-EDTA, and other routine cell culture reagents were purchased from Biological Industries (Kibbutz Beit Haemek, Israel) and Invitrogen-Gibco (Grand Island, NE, USA). H_2_O_2_ was purchased from Sigma-Aldrich (St. Louis, MO, USA). The CCK8 kit was provided by TransGen Biotech (Beijing, China). The ROS and LDH assay kits were offered by Beyotime Biotech (Yancheng, China). The Annexin V-FITC/PI apoptosis detection kit was acquired from BD Bioosciences (Franklin Lakes, NJ, USA). The JC-1 mitochondrial membrane potential kit was provided by Yeasen Biotech (Shanghai, China). DAPI solution (ready-to-use) was supplied by Solarbio Biotech (Beijing, China). The ROS, MDA, CAT, and SOD detection kits were provided by Jingmei Biotechnology Company (Yancheng, China). The following antibodies: bax, bcl-2, cleaved PARP, cleaved caspase3, Phospho-p38, Phospho-Erk, Phospho-JNK, GAPDH, β-tubulin, and antibody cytochrome c, were purchased from Cell Signaling Technology (Danvers, MA, USA).

### 3.2. Extraction and Purification of the Polysaccharide

The dried *P. sajor-caju* powders were extracted under the condition of 30:1 liquid to material ratio at 80 °C for 2 h. The extracting solution was concentrated to one fourth of its original volume and then precipitated by adding ethanol to a final concentration of 75% (*v*/*v*) overnight at 4 °C. This method is used to remove proteins from polysaccharides, and the oligosaccharides and other small molecules were removed by dialysis. Then, the polysaccharide samples were freeze dried, and named PSP for further purification.

The PSP was dissolved in deionized water and filtered with a 0.22 μm microporous membrane, and then applied to a DEAE FF column (1.6 × 10 cm), followed by a linear gradient elution using 0 to 1 mol/L of sodium chloride at 1.0 mL/min. The eluted fractions were collected by automatic collector. Then, the main fraction PSP2 was further purified using the Sephacry S-400 HR column (1.6 × 60 cm) and the main eluted fraction was dialyzed and lyophilized, namely PSP2-1. Furthermore, the carbohydrate and protein contents of PSP2-1 were determined using the phenol-sulfuric acid method and Bradford method [54,55], respectively.

### 3.3. Molecular Weight and Monosaccharide Composition

The molecular weight of PSP2-1 was determined by GPS-RI-MALLS [56]. In brief, 5 mg of the PSP2-1 polysaccharide sample was dissolved in 1 mL NaNO_3_ (0.1 mol/L) for 20 min at 100 °C. The mixture was centrifuged at 14,000 rpm for 10 min, and 100 μL of supernatant was measured by GPS-RI-MALLS (DAWN HELEOSII, Wyatt Technology, Santa Barbara, CA, USA). The detecting system included an Agilent 1260 HPLC system (Agilent, Palo Alto, Santa Clara, CA, USA) and an Optilab T-rEX refractive index detector (Wyatt technology, CA, USA), and the analytic column consisted of Ohpak SB-805 HQ, Ohpak SB-804 HQ, and Ohpak SB-803 HQ (Shodex, Asahipak, Tokyo, Japan). The mobile phase was 0.1 mol/L of NaNO_3_ solution at a flow rate of 0.4 mL/min, and the temperature of the column was maintained at 60 °C. The sample data were analyzed with ASTRA6.1 software (Wyatt Technology, Santa Barbara, CA, USA).

The PSP2-1 polysaccharide sample (5 mg) was hydrolyzed with 2.5 mol/L trifluoroacetic acid for 120 min at 121 °C. The hydrolysate was dried by nitrogen, then washed two times with methanol. Furthermore, the sample was dissolved with sterile water, then was analyzed by ion chromatography (Thermo Fisher Scientific, 81 Wyman Street, Waltham, MA, 02454, ICS5000) with CarboPac™ PA20 columns (Dionex, Sunnyvale, CA, USA, 4 × 250 mm) combined with pulsed amperometric detection. The monosaccharide composition was analyzed according to the retention time of a standard monosaccharide.

### 3.4. FTIR and UV Spectra Analysis

A total of 2 mg of dried PSP2-1 was mixed with potassium bromide (KBr) powder (150 mg), ground, and then pressed into a disc for an FTIR spectrometer (Nicolet 5700, Thermoscientific, Waltham, MA, USA) determination in the range of 400–4000 cm^−1^. UV spectra of PSP2-1 (1 mg) was recorded with a UV spectrometer (Genesys 180, Thermoscientific, USA) from 190 to 800 nm.

### 3.5. Methylation of PSP2-1

The methylation analysis of PSP2-1 was determined based on the previously reported method [57]. A total of 5 mg of the polysaccharide sample PSP2-1 was dissolved in dimethyl sulfoxide, and 20 mg of sodium hydroxide was added for 1 h. Then, 10 μL iodomethane (CH_3_I) and 500 μL dichloromethane were added for 20 min, respectively, after being centrifuged, discarded after the aqueous phase, and the dichloromethane was evaporated. Furthermore, the sample was hydrolyzed with 2.5 mol/L trifluoroacetic acid at 121 °C for 90 min, Then, 50 μL of 2 mol/L ammonia and 50 μL of 1 mol/L NaBD_4_ was added for 2.5 h at room temperature, and acetic acid was added to terminate the reaction. Lastly, the sample was acetylated by adding 250 μL of acetic anhydride for 2.5 h at 100 °C, and the methylated alditol acetate derivatives were analyzed with an HP-5 ms quartz capillary column (30 m × 0.25 mm× 0.25 μm, Agilent Technology, USA), which was connected to the GC-MS system (GC7890A-MS5975C, Agilent Technology, USA).

### 3.6. Cell Culture

The mouse hippocampal neuron cells (HT22 cells) were cultured in Dulbecco’s modified Eagle’s medium with 10% fetal bovine serum, 1% biomyc-3, 100 units/mL penicillin, and 100 units/mL streptomycin at 37 °C in an atmosphere containing 5% CO_2_.

### 3.7. Cell Viability Assay

HT22 cells were added into the 96-well plates at 2 × 10^4^ cells/well and pre-treated with PSP2-1 (0, 50, 100, to 150 μg/mL) for 24 h. Next, 500 μM of H_2_O_2_ were added and co-cultured with PSP2-1 for another 2 h. Then, the supernatants were discarded and 20 μL CCK8 solution (20 μL CCK8:180 μL medium) was added to each well. After 1 h, the absorbance was read at 450 nm by an ELISA microplate reader.

### 3.8. Detection of Intracellular ROS Level

The intracellular ROS levels were detected by flow cytometry through an oxidation-sensitive fluorescent probe (DCFH-DA). HT22 cells were pre-treated with PSP2-1 (0, 50, 100, and 150 μg/mL) for 24 h in 6-well plates and co-cultured with 500 μM of H_2_O_2_ for 2 h. According to the instructions, 2′-7′-dichlorodihydrofluorescein diacetate (DCFH-DA) was diluted with DMEM without FBS and added to the cell of plates for 20 min in the dark at 37 °C. Next, cells were washed with PBS three times and detected by flow cytometry.

### 3.9. Apoptosis Quantification

The HT22 cells apoptosis percentage was evaluated by flow cytometry analysis using Annexin V-FITC and PI fluorescence. The cells were seeded in 12-well plates and treated with different concentrations of PSP2-1 (0, 50, 100, and 150 μg/mL) for 24 h and 500 μM H_2_O_2_ for 2 h. Then, the medium was removed by centrifuge, washed three times with a binding buffer, and stained with Annexin V-FITC and PI with a binding buffer. Next, the cells were collected and centrifuged at 1500 rpm for 5 min at 4 °C, and analyzed using flow cytometry.

### 3.10. Lactate Dehydrogenase (LDH) Release Analysis

After the treatment by PSP2-1 (0, 50, 100, and 150 μg/mL) for 24 h and 500 μM H_2_O_2_ for 2 h of HT22 cells in 96-well plates, the 96-well plates were centrifuged at 400× *g* for 5 min. Next, according to the protocol, the supernatant was used by the LDH analysis kit and the release of LDH in the culture medium was determined.

### 3.11. DAPI Staining

After the HT22 cells were pre-treated with PSP2-1 (0, 50, 100, and 150 μg/mL) for 24 h and co-cultured with 500 μM H_2_O_2_ for 2 h, the cells were washed with PBS three times and fixed with 4% paraformaldehyde for 20 min. After washing them three times with PBS, they were stained with 4′,6-diamidino-2-phenylindole (DAPI) for 10 min and analyzed with a fluorescence microscope.

### 3.12. JC-1 Assay for Mitochondrial Membrane Potential

JC-1 staining was used to detect mitochondrial membrane potential (MMP). After HT22 cells were grown in 24-well plates for 24 h, the cells were pretreated with different doses of PSP2-1 (0, 50, 100, and 150 μg/mL) for 24 h, followed by exposure to 500 μM H_2_O_2_ for 2 h. Then, the cells were washed with PBS three times for 5 min each time, and incubated with JC-1 dyes at 37 °C in the dark for 15 min. Finally, the cells were washed with PBS three times for 5 min each time, and the intracellular fluorescence was analyzed by flow cytometry.

### 3.13. Immunohistochemistry

HT22 cells were seeded onto 48-well plates, which were pre-treated by PSP2-1 (0, 50, 100, and 150 μg/mL), for 24 h and co-cultured with 500 μM H_2_O_2_ for 2 h and fixed with 4% paraformaldehyde for 30 min. Subsequently, each well was washed by PBS three times for 5 min each time and penetrated by 0.5% Triton X-100 for 20 min. After washing them three times with PBS, each well was incubated in a blocking medium (5% bovine serum albumin) for 30 min at room temperature and cytochrome c primary antibody was added overnight at 4 °C. Then, washing them three times with PBS, the cells were incubated with secondary Alexa Flour 488-goat anti-rabbit antibody for 1 h at room temperature and stained with DAPI for 10 min. Finally, the cells were analyzed with a fluorescence microscope.

### 3.14. Western Blot Analysis

After the HT22 cells were pre-treated with PSP2-1 (0, 50, 100, and 150 μg/mL) for 24 h and co-cultured with 500 μM H_2_O_2_ for 2 h. The cells were lysed in a RIPA buffer with 1% protease and phosphatase inhibitors for 30 min. A BCA protein assay kit was used for the determination of total protein concentrations. Then, an equal amount of protein (20 μg) for each sample was separated by 15% SDS-PAGE and transferred onto polyvinylidene difluoride PVDF membranes. Then, the membranes were blocked with 5% bovine serum albumin (BSA) for 1 h on the horizontal shaker at room temperature, followed by an overnight incubation at 4 °C of the primary antibodies, including bcl-2, bax, P-p38, P-JNK, P-Erk, cleaved caspase-3, cleaved PARP, GAPDH, and β-tubulin. The next day, after washing with TBST three times for 10 min each time, the membranes were incubated with horseradish peroxidase (HRP)-conjugated secondary antibody for 1 h at room temperature after washing with TBST three times for 10 min each time. The protein bands were visualized by an ECL kit and quantified by using Image J software.

### 3.15. Establishment of Animal Model

Eight-week-old SPF grade BALB/c mice were purchased from Wei tong Li hua Biotechnology Co., Ltd. A total of 50 mice, half male and half female, were randomly divided into 5 groups, the control group (saline), the model group (D-galactose 200 mg/kg), PSP2-1 low dose group (100 mg/kg), PSP2-1 medium dose group (200 mg/kg), and PSP2-1 high dose group (400 mg/kg). In the first 49 days, D-galactose was injected into each group except the control group every day; from day 50, in addition to daily intraperitoneal injection of D-galactose, the low dose group was given PSP2-1 (100 mg/kg), the medium dose group was given PSP2-1 (200 mg/kg), and the high dose group was given PSP2-1 (400 mg/kg) by gavage every day for 42 days [58]. Then, the water maze experiment was conducted and the oxidative indexes in the serum and brains of mice were detected.

### 3.16. Morris Water Maze Test (MWM)

The water maze experiment was carried out for 6 days. The mice were trained in the first 5 days. Each mouse was put into the pool from four different quadrants for 120 s to find the hidden platform. If the mouse did not find the platform, it was placed on the hidden platform for 15 s after the end, so that the mice could remember the location of the platform. After 5 days of training, the formal test began by performing a 120 s detection experiment on the mice, and recording the escape latency of finding the hidden platform during this period. The last day of the test was divided into two experiments: the first experiment was the no platform test, where we removed the platform, and investigated the number of times that the mice found the platform position within the specified time (60 s); the second experiment was to have the platform test to investigate the time taken by the mice to find the platform.

### 3.17. Oxidative Stress Index Detection

Blood was collected from mice, and then centrifuged (3500 rpm, 10 min) for 30 min. Serum was separated and stored at 4 °C for later use. The brain tissues of mice were homogenized in cold physiological saline, and centrifuged (10,000 rpm, 10 min) at 4 °C. The supernatant was collected for biochemical analysis. Levels of MDA, SOD, ROS and CAT in the brain tissue and serum were measured by detection with a microplate reader using the corresponding commercial assay kits (Jingmei Biotechnology Co., Ltd., Yancheng, China). All procedures were performed in accordance with the manufacturer’s instructions.

### 3.18. Statistical Analysis

All of the numerical experiments data are expressed as mean ± standard deviation (SD). The statistical analyses were carried out with the SPSS 17.0 software package.

## 4. Conclusions

In summary, we successfully isolated and purified a novel polysaccharide PSP2-1 with a molecular weight of 44.9 kDa from the fruiting body of *P. sajor-caju*. Furthermore, the PSP2-1 is mainly composed of fucose, galactose, glucose, and mannose with T-Fuc, 1,6-Gal, T-Glc, 1,6-Glc, 1,3,6-Glc, 1,3-Man, 1,2,6-Man, and T-Man glycosidic bonds. Finally, the polysaccharide PSP2-1 can protect nerve cells from oxidative damage and apoptosis induced by hydrogen peroxide by regulating the MAPK signaling pathway. It is also able to improve the learning and memory ability of mice with aging induced by D-galactose, and improve the degree of oxidative damage.

## Figures and Tables

**Figure 1 nutrients-14-04047-f001:**
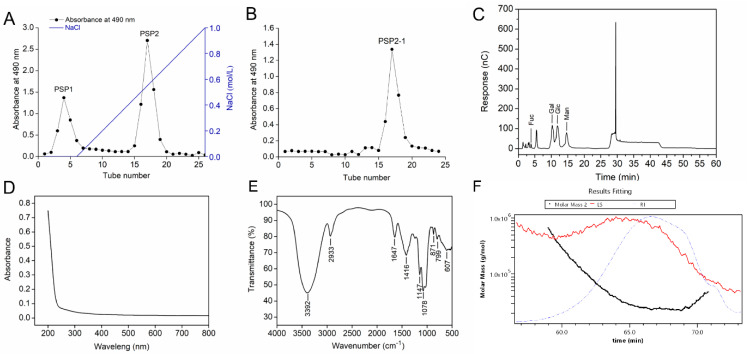
Purification and physicochemical properties of polysaccharide from *P. sajor-caju*. (**A**) Elution curve of PSP2 from DEAE FF; (**B**) Elution curve of PSP2-1 from Sephacry S-400 HR; (**C**) Monosaccharide composition of PSP2-1; (**D**) UV spectra of PSP2-1; (**E**) FTIR spectra of PSP2-1; (**F**) Molecular weight distribution of PSP2-1.

**Figure 2 nutrients-14-04047-f002:**
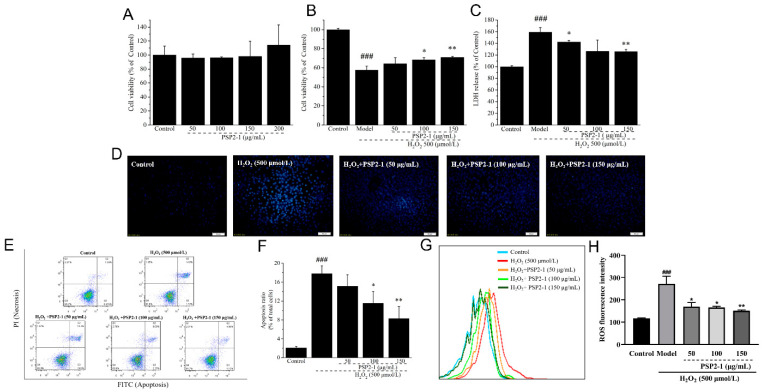
(**A**) Effects of PSP2-1 on the cell viability of HT22 cells; (**B**) Effects of PSP2-1 on the cell viability of H_2_O_2_-treated HT22 cells; (**C**) Effects of PSP2-1 on the LDH release of H_2_O_2_-treated HT22 cells; (**D**) Effects of PSP2-1 on the morphological characteristics of H_2_O_2_-treated HT22 cells measured by 4′,6-diamidino-2-phenylindole (DAPI) staining under fluorescence microscope (Scale bar = 50 μm); (**E**) Dot plots of the effect of PSP2-1 on H_2_O_2_-induced apoptosis in HT22 cells assessed by flow cytometry; (**F**) The quantification histogram of the percentage of apoptotic cells in total cell population; (**G**) Effect of PSP2-1 on ROS generation of H_2_O_2_-treated HT22 cells measured by flow cytometry; (**H**) Quantitative data of ROS generation in HT22 cells treated with H_2_O_2_ by PSP2-1. All of the data were analyzed using a one-way ANOVA and they are expressed as means ± standard deviation (SD). ### *p* < 0.001 in a comparison with the control group; * *p* < 0.05 and ** *p* < 0.01 as compared with the model group.

**Figure 3 nutrients-14-04047-f003:**
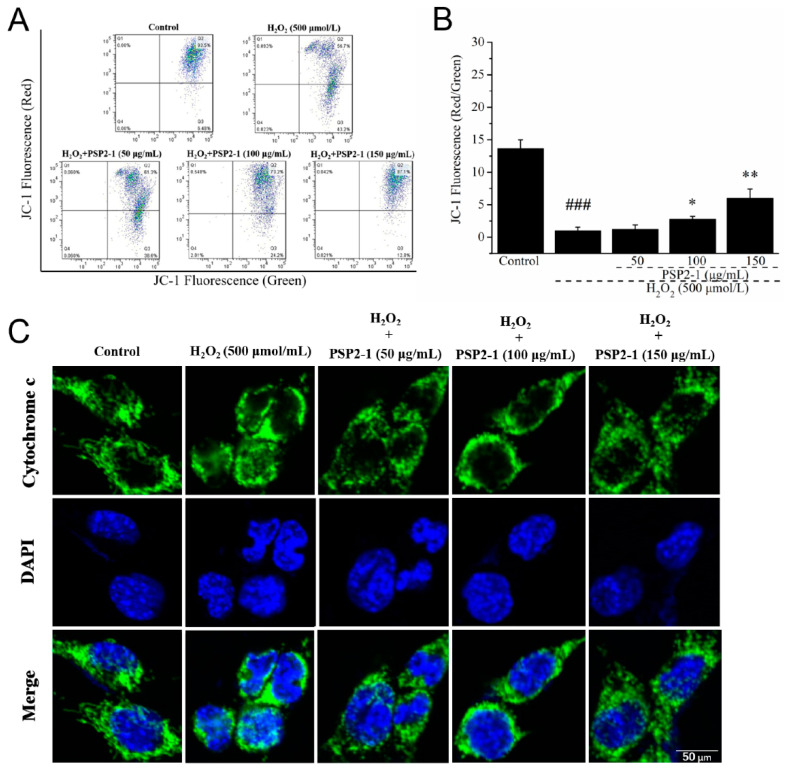
(**A**) Effects of PSP2-1 on mitochondrial membrane potential change in H_2_O_2_-treated HT22 cells measured by JC-1 staining on flow cytometry; (**B**) Quantification histogram of the percentage of red/green fluorescence; (**C**) Effects of PSP2-1 on the cytochrome c release of H_2_O_2_-treated HT22 cells measured by laser confocal microscope (Scale bar = 50 μm). All of the data were analyzed using a one-way ANOVA and they are expressed as means ± SD. ### *p* < 0.001 in a comparison with the control group; * *p* < 0.05 and ** *p* < 0.01 as compared with the model group.

**Figure 4 nutrients-14-04047-f004:**
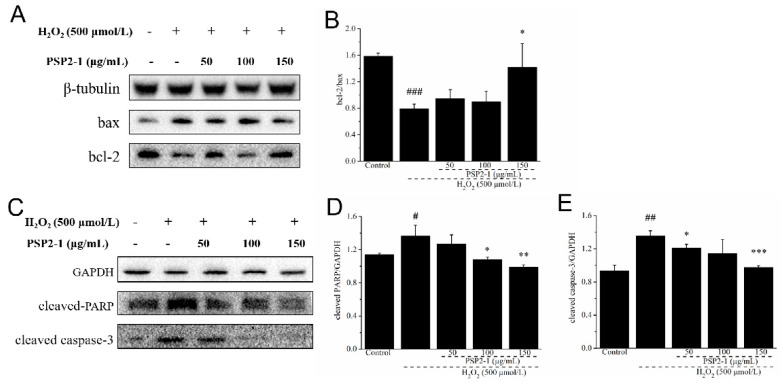
(**A**) Effect of PSP2-1 on the protein expression of bax and bcl-2 in H_2_O_2_-treated HT22 cells measured by Western blot; (**B**) Quantitated results of the expression ratio of bax and bcl-2; (**C**) Effect of PSP2-1 on the protein expression of cleaved PARP and cleaved caspase-3 in H_2_O_2_-treated HT22 cells measured by Western blot; (**D**,**E**) Quantitated results of cleaved PARP and cleaved caspase-3 protein levels. All of the data were analyzed using a one-way ANOVA and they are expressed as means ± SD. # *p* < 0.05, ## *p* < 0.01 and ### *p* < 0.001 in a comparison with the control group; * *p* < 0.05, ** *p* < 0.01 and *** *p* < 0.001 as compared with the model group.

**Figure 5 nutrients-14-04047-f005:**
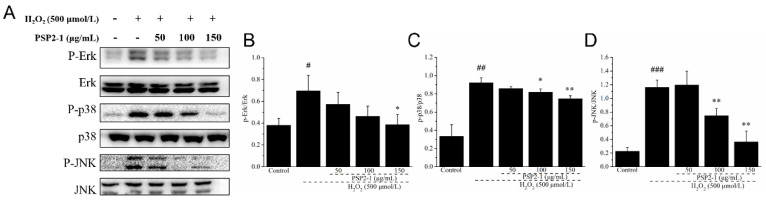
Effect of PSP2-1 on the MAPK signaling pathway in H_2_O_2_-treated HT22 cells. (**A**) Effect of PSP2-1 on protein expression of P-Erk, Erk, P-p38, p38, P-JNK, and JNK in H_2_O_2_-treated HT22 cells measured by Western blot; (**B**–**D**) Quantitated results of relative expression P-Erk, P-p38, and P-JNK. The data were analyzed using a one-way ANOVA and they are expressed as means ± SD. # *p* < 0.05, ## *p* < 0.01 and ### *p* < 0.001 in a comparison with the control group; * *p* < 0.05 and ** *p* < 0.01 as compared with the model group.

**Figure 6 nutrients-14-04047-f006:**
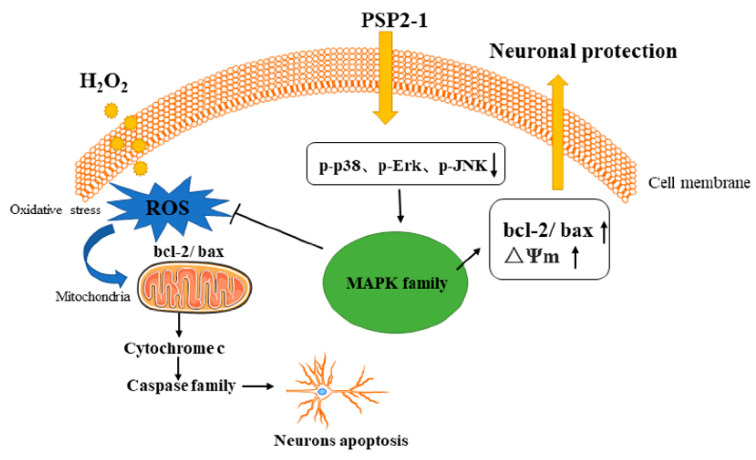
Overview of the mechanism pathways of PSP2-1 inhibiting H_2_O_2_-induced apoptosis of oxidatively damaged nerve cells.

**Figure 7 nutrients-14-04047-f007:**
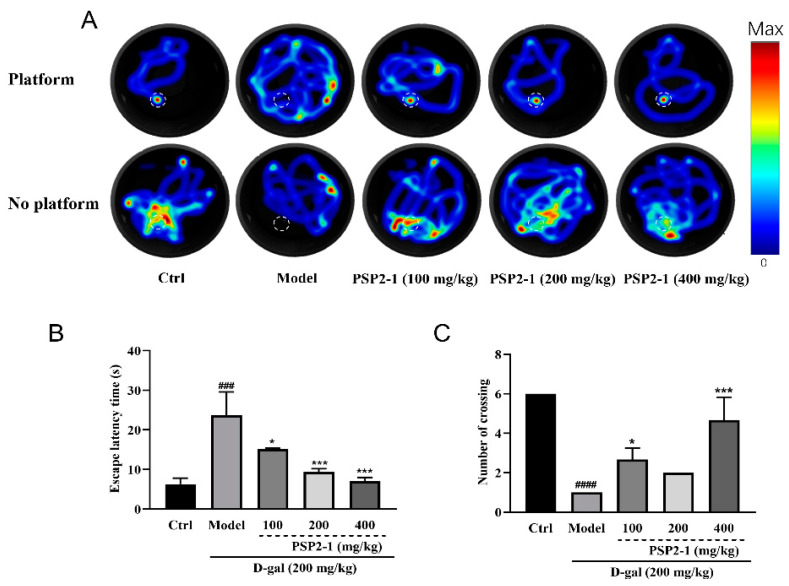
Effects of PSP2-1 on learning and memory ability of mice with aging induced by D-galactose. (**A**) Thermal infrared tracks of different groups of mice in the water maze; (**B**) Escape latency time in different groups of mice; (**C**) The number of mice in different groups looking for a platform when there is no platform. All of the data were analyzed using a one-way ANOVA and they are expressed as means ± SD. ### *p* < 0.001 and #### *p* < 0.0001 in a comparison with the control group; * *p* < 0.05 and *** *p* < 0.001 as compared with the model group.

**Figure 8 nutrients-14-04047-f008:**
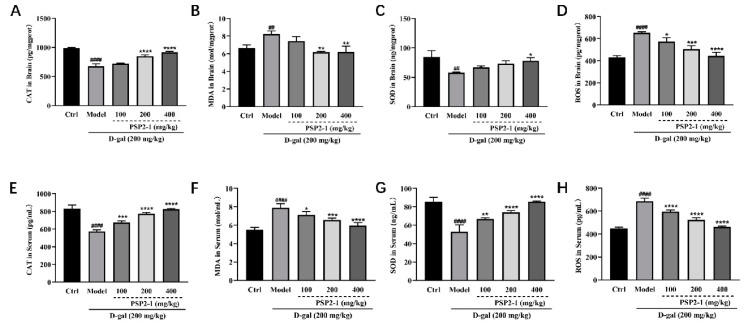
Effect of PSP2-1 on oxidative stress injury induced by D-galactose in aging mice. (**A**) CAT levels in the brains of different groups of mice; (**B**) MDA levels in the brains of different groups of mice; (**C**) SOD levels in the brains of different groups of mice; (**D**) ROS levels in the brains of different groups of mice; (**E**) CAT levels in the serum of different groups of mice; (**F**) MDA levels in the serum of different groups of mice; (**G**) SOD levels in the serum of different groups of mice; (**H**) ROS levels in the serum of different groups of mice. All of the data were analyzed using a one-way ANOVA and they are expressed as means ± SD. ## *p* < 0.01 and #### *p* < 0.0001 in a comparison with the control group; * *p* < 0.05, ** *p* < 0.01, *** *p* < 0.001 and **** *p* < 0.0001 as compared with the model group.

**Table 1 nutrients-14-04047-t001:** Composition and physicochemical characteristic of PSP2-1.

Sample	PSP2-1
Carbohydrate (%)	85.30
Protein (%)	3.10
Molecular weight (kDa)	44.90
Monosaccharide composition (mol %)	
Fucose (mol %)	2.40
Galactose (mol %)	28.81
Glucose (mol %)	30.63
Mannose (mol %)	37.79

**Table 2 nutrients-14-04047-t002:** Results of the methylation analysis of PSP2-1.

Methylation	Retention Time (min)	Linkage Type	Molar Ratio (%)
1,5-di-O-acetyl-6-deoxy-2,3,4- tri-O-methyl fucitol	11.03	T-Fuc (p)	1.51
1,5,6-tri-O-acetyl-2,3,4-tri-O- methyl galactitol	18.80	1,6-Gal (p)	25.57
1,5-di-O-acetyl-2,3,4,6-tetra-O- methyl glucitol	14.21	T-Glc (p)	19.57
1,5,6-tri-O-acetyl-2,3,4-tri-O- methyl glucitol	17.79	1,6-Glc (p)	4.33
1,3,5,6-tetra-O-acetyl-2,4-di-O- methyl glucitol	20.71	1,3,6-Glc (p)	6.74
1,3,5-tri-O-acetyl-2,4,6-tri-O- methyl mannitol	16.96	1,3-Man (p)	13.44
1,2,5,6-tetra-O-acetyl-3,4-di-O- methyl mannitol	21.49	1,2,6-Man (p)	16.72
1,5-di-O-acetyl-2,3,4,6-tetra-O- methyl mannitol	14.13	T-Man (p)	7.16

## Data Availability

All data used in this study are available from the corresponding authors upon reasonable request.

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
