# Peer review of "Structural Characterization of a Pleurotus sajor-caju Polysaccharide and Its Neuroprotection Related to the Inhibition of Oxidative Stress"

_nutrients, 2022, doi:10.3390/nu14194047_

Round 1

Reviewer 1 Report

I appreciate the author presenting this research article emphasizing the  neuroprotecitve effects of PSP is related to inhibition on oxidative stressThis is a well-designed and organized research article. The results support partly the hypothesis. My comments are as follows,

1.  The anti-oxidative stress effects in animal study were not shown in current study, please provide. 

2.  Please provide the limitations of current study

Reviewer 2 Report

In the present study, the polysaccharide PSP2-1 has been isolated and purified from Pleurotus sajor-caju and assessed for its neuroprotective properties. Particularly, the Authors highlighted the ability of PSP2-1 to counteract the injury induced by H2O2 in mouse hippocampal neuron HT22 cell lines. To this end, multiple parameters, including cell viability, LDH release, apoptosis rate, oxidative stress, morphology, mitochondrial membrane potential and related markers were measured. The obtained results were consistent with protective effects of PSP2-1 towards H2O2 likely mediated by the modulation of MAPK signaling pathway. The substance also improved the learning and memory ability of aging mice induced by D-galactose thus suggesting a future nutraceutical interest in this substance and in Pleurotus sajor-caju.

Despite this interest, some issues required to be improved in order to make the manuscript suitable for publication.

Particularly, data require to be better discussed respect to other similar compounds or other studies focused on Pleurotus sajor-caju and related fungi. Indeed, results of in vivo evaluation are scantily discussed. Moreover, the study design, the choice of the selected models, especially the animal model, and the impact of the obtained results should be highlighted.

Other issues to be considered are listed below.

-          Line 63: delete “in animal experiments.”

-          Line 66: “P. sajor-caju” should be italicized

-          Al the Figures, especially Figures 1-5 and subfigures, require to be resized and readability improved.

-          Table 1: the unit of measurement (%?) for fucose, galactose, glucose and mannose should be added

-          Lines 92-94 “Methylation detection is an important means to analysis the glycoside bond of poly-92 saccharide. The methylated derivatives of PSP2-1 were determined by GC-MS. Table 2 93 showed that the PSP2-1 showed eight main components,”: check and correct

-          Line 104: the choice of the H2O2 concentration along with time exposure should be reported.

-          Line 104 “cell model of AD”: this wording should be supported by literature. The abbreviation should be explained. If the Authors aimed at selecting a model of AD, it should be of interest to evaluate the protective effects of PSP2-1 against damage induced by Aβ protein.

-          Line 105 “on H2O2-induced HT22 cell”: “towards damage induced by H2O2 in HT22 cells”.

-          Lines 106-107: explain the choice of the concentration range.

-          Figure 2: considering the lacking cytotoxicity occurring at the concentration of 200 μg/mL, it could be interesting to study the concentration-response trend of the cytoprotective effect of PSP2-1.

-          Figure 2G: Quantification of oxidative stress should be included.

-          Paragraphs 2.5, 2.6, 2.7 and 2.7: results should be detailed.

-          Original western blotting membranes should be provided as supporting materials.

-          Data discussion with respect to literature on fungi and other polysaccharides is scanty.

-          Line 271: identification of the species should be added.

Round 2

Reviewer 2 Report

The Authors addressed all my concerns.

Author Response

Thank you very much